# Comparative Evaluation of Bleomycin- and Collagen-V-Induced Models of Systemic Sclerosis: Insights into Fibrosis and Autoimmunity for Translational Research

**DOI:** 10.3390/ijms26062618

**Published:** 2025-03-14

**Authors:** Lőrinc Nagy, Gábor Nagy, Tamás Juhász, Csaba Fillér, Gabriella Szűcs, Zoltán Szekanecz, György Vereb, Péter Antal-Szalmás, Árpád Szöőr

**Affiliations:** 1Department of Biophysics and Cell Biology, Faculty of Medicine, University of Debrecen, H-4032 Debrecen, Hungary; nagy.lorinc@med.unideb.hu (L.N.);; 2Department of Laboratory Medicine, Faculty of Medicine, University of Debrecen, H-4032 Debrecen, Hungary; 3Department of Anatomy, Histology and Embryology, Faculty of Medicine, University of Debrecen, H-4032 Debrecen, Hungary; juhaszt@anat.med.unideb.hu (T.J.);; 4Department of Rheumatology and Immunology, Faculty of Medicine, University of Debrecen, H-4032 Debrecen, Hungary; szucsgpafi@gmail.com (G.S.);; 5HUN-REN-UD Cell Biology and Signaling Research Group, Faculty of Medicine, University of Debrecen, H-4032 Debrecen, Hungary; 6Faculty of Pharmacy, University of Debrecen, H-4032 Debrecen, Hungary

**Keywords:** systemic sclerosis, fibrosis, autoimmunity, bleomycin model, collagen-V model

## Abstract

Systemic sclerosis (SSc) is a complex autoimmune disease characterized by fibrosis, immune dysregulation, and vascular dysfunction, yet its pathogenesis remains incompletely understood. This study compares two widely used animal models of SSc—the bleomycin-induced fibrosis model and the collagen-V-induced autoimmune model—to evaluate their ability to replicate key disease features. In the bleomycin model, consistent cardiac fibrosis was observed across treatment groups despite variability in fibrosis in the skin and lungs, suggesting organ-specific differences in susceptibility. The collagen-V model demonstrated robust autoantibody production against collagen-V, confirming its utility in studying immune activation, though fibrosis was largely confined to the heart. While the bleomycin model excels at mimicking rapid fibrosis and is suitable for testing antifibrotic therapies, the collagen-V model provides insights into antigen-specific autoimmunity. Both models highlight the dynamic nature of fibrosis, where ECM deposition and degradation occur concurrently, complicating its use as a quantitative disease marker. Cardiac fibrosis emerged as a consistent feature in both models, emphasizing its relevance in SSc pathophysiology. Combining these models or refining their design through hybrid approaches, extended timelines, or sex and age adjustments could enhance their translational utility. These findings advance understanding of SSc mechanisms and inform therapeutic development for this challenging disease.

## 1. Introduction

Systemic sclerosis (SSc) is a chronic autoimmune disorder characterized by excessive fibrosis, vascular dysfunction, and immune dysregulation, which collectively contribute to significant morbidity and mortality rates [1]. The disease affects the skin, internal organs, and vasculature, presenting a complex clinical phenotype. While the precise mechanisms underlying SSc remain incompletely understood, autoimmunity, fibrosis, and vascular dysfunction are widely recognized as hallmark features [2]. Efforts to develop effective treatments are complicated by the clinical and pathological heterogeneity of the disease and the lack of universally accepted animal models that fully simulate its multifactorial nature [3].

Despite these challenges, in vivo models are essential for understanding the pathophysiology and for testing potential therapeutic strategies. Among the most widely used models is Yamamoto’s bleomycin-induced fibrosis model, which involves the local or systemic administration of bleomycin, a chemotherapeutic agent [4]. This model reliably induces cutaneous fibrosis, mimicking key features of SSc such as excessive collagen deposition, fibroblast activation, and immune cell infiltration. However, the bleomycin-induced model is inherently limited, as it lacks the autoimmune component central to SSc. The fibrosis induced by bleomycin arises from a direct chemical insult rather than immune dysregulation, which is the driving force of SSc in patients [5]. The tissue damage is primarily driven by oxidative stress, leading to a biphasic response that mirrors systemic sclerosis (SSc) aspects. The initial inflammatory phase involves immune cell infiltration and the release of cytokines such as TNF-α, IL-6, and TGF-β, contributing to endothelial injury and fibroblast activation [6]. This progresses to a chronic fibrotic phase, characterized by excessive fibroblast proliferation, myofibroblast differentiation, and extracellular matrix (ECM) deposition, primarily of collagen I and III, leading to progressive tissue stiffening [7]. Oxidative stress and persistent cytokine signaling sustain this profibrotic environment, resembling SSc pathogenesis, where vascular and immune dysfunction drive fibrosis in the skin and internal organs [8] (Figure 1 orange boxes). While valuable for studying fibrotic pathways, this model does not fully capture the multifactorial pathogenesis of SSc [3]. To address some of these limitations, genetic modifications have been explored to enhance the bleomycin-induced SSc phenotype. Fli1-haploinsufficiency has been shown to exacerbate bleomycin-induced fibrosis, leading to a more severe SSc-like phenotype with enhanced vascular dysfunction and immune abnormalities [9].

While the bleomycin model effectively induces fibrosis, its reliance on direct chemical insult limits its ability to fully replicate SSc pathogenesis. HOCl, a reactive oxygen species produced by myeloperoxidase in neutrophils, triggers endothelial injury, immune activation, and fibroblast proliferation [10]. Its systemic administration leads to vascular inflammation, closely mimicking key pathological features of SSc [11]. However, fibrosis in this model is less consistent and develops more gradually compared to the bleomycin model, making it less suitable for fibrosis-targeted therapeutic evaluations. Given our focus on well-established fibrotic processes, we prioritized the bleomycin model for its reproducibility and translational relevance.

In contrast, antibody-based models, such as those employing collagen-V immunization with adjuvants, are designed to replicate the autoimmune aspects of the disease [12]. They are primarily driven by adaptive immune activation, where dendritic cells (APCs) present collagen antigens, leading to Th1 and Th17 cell responses and the secretion of TNF-α, IFN-γ, and IL-17, which drive inflammation (Figure 1, blue boxes). These models induce the production of autoantibodies that are diagnostic hallmarks of SSc, including antinuclear antibodies (ANAs) and anti-topoisomerase I antibodies [13,14]. While these autoantibodies are pivotal in diagnosing SSc, their precise roles in disease pathogenesis remain unclear. Antibody-based models thus provide a valuable platform to investigate the immune components of SSc, but their ability to mimic fibrotic and vascular aspects of the disease is less well-defined [15]. It is important to highlight that macrophages and fibroblasts (Figure 1; green-dashed box) are shared components in both models, highlighting their central role in fibrotic progression.

Despite their strengths, a comprehensive comparison of bleomycin-induced and antibody-based models of SSc has yet to be conducted. Such a comparison is essential to assess the strengths, limitations, and translational relevance of these models. Furthermore, these models offer a unique opportunity to explore the broader pathophysiological aspects of SSc, including the roles of genetic predisposition, epigenetic regulation, and environmental triggers. The assessment of antinuclear (ANA) and anticytoplasmic autoantibodies using the HEp-2 IFA experiment was a key focus of our study, given the well-established role of autoantibodies in systemic sclerosis (SSc) pathogenesis. Autoantibodies such as anti-topoisomerase I, anti-centromere, and anti-RNA polymerase III have been strongly associated with disease severity, specific clinical manifestations, and survival outcomes in SSc patients [16]. Given these clinical implications, our study aimed to determine whether ANA and anticytoplasmic autoantibodies influence disease progression and survival outcomes in our murine models. Notably, the relationship between autoantibodies and disease pathogenesis remains an area of active investigation, with questions persisting about whether these markers are merely diagnostic or directly contribute to disease progression [17].

This study aims to address these critical gaps by systematically comparing bleomycin-induced and collagen-V/adjuvant-based models of SSc. We evaluate the applicability of each of these models to specific research objectives and their respective strengths and limitations. Additionally, we explore the implications of autoantibody production, examining whether these markers are solely diagnostic or play an active role in the pathophysiology of SSc.

## 2. Results

### 2.1. Bleomycin and Simultaneous Treatment with Lymphocytes from SSc Patients Accelerates Mortality and Impacts Collagen Synthesis

The bleomycin-induced animal model is a widely used approach to study SSc due to its ability to mimic key fibrotic processes, including excessive collagen deposition and fibroblast activation. Our study utilized this model to evaluate the combined effects of bleomycin and treatment with peripheral blood mononuclear cells (PBMC) from SSc patients, explicitly focusing on survival outcomes and collagen synthesis [4].

Survival outcomes varied significantly across the treatment groups (Figure 2A). Animals in the bleomycin and SSc-derived PBMC co-treated (SSc-PBMC + bleomycin) group exhibited the shortest survival, with all animals succumbing by day 29 and a median survival time of 23 days. Similarly, groups treated with “PBMC + Bleomycin” and “Bleomycin” alone displayed median survival times between days 32–40 and 42–44, respectively. In contrast, mice treated with healthy donor-derived PBMCs demonstrated a prolonged lifespan, surviving an average of 76–78 days. Notably, both the non-treated (“NT”) group and the “SSc-PBMC” group, which received SSc donor-derived PBMCs alone, survived for the entire duration of the 81-day experiment. Histological analysis is crucial for assessing tissue-level changes associated with SSc models, particularly regarding fibrosis and extracellular matrix (ECM) remodeling. Initial hematoxylin-eosin staining evaluation revealed no significant structural differences between treatment groups (Appendix A). Similarly, Masson’s trichrome staining, which highlights connective tissue, did not show notable differences. In contrast, Picrosirius Red staining, which identifies collagen deposition, highlighted substantial differences (Figure 2B,C; Appendix A). A general reduction in collagen content was observed across most organ samples from treated groups, as indicated by decreased pixel intensities compared to the non-treated control group (Appendix A). However, the heart was an exception, showing increased collagen deposition in all treatment groups. The “SSc-PBMC + Bleomycin” and “Bleomycin” groups exhibited significant increases in collagen content in the heart, aligning with their poor survival outcomes. Interestingly, while the “PBMC + Bleomycin” group showed survival comparable to that of the “SSc-PBMC + Bleomycin” and “Bleomycin” groups, its increase in collagen deposition was less pronounced. This discrepancy may suggest that other factors, such as systemic inflammation or additional molecular mechanisms induced by PBMCs, contributed to the rapid mortality in this group. These findings indicate a complex interplay between collagen deposition, immune responses, and survival outcomes, emphasizing the need for further investigation into the mechanisms underlying these results.

### 2.2. Antibody Production in the Bleomycin-Induced SSc Model Fails to Explain Survival Outcomes

Autoantibody production is a hallmark of SSc and has often been used as a diagnostic and prognostic marker. The HEp-2 immunofluorescence assay (IFA), a valuable tool for detecting and characterizing autoantibodies, can provide insights into immune dysregulation and potential disease mechanisms. In this study, the HEp-2 IFA experiment aimed to assess antinuclear (ANA) and anticytoplasmic autoantibodies across treatment groups to investigate their possible role in survival outcomes and disease progression.

The results revealed that all treated groups, except the non-treated (“NT”) and “Bleomycin” groups, produced human IgG antibodies, which demonstrated the survival and engraftment of human T and B cells in the recipient mice. The absence of human and mouse antibodies in the NT and Bleomycin groups can be attributed to the lack of both mouse and human B cells in these animals. Interestingly, based on the fluorescence pattern, none of the detected antibodies targeted the homolog autoantigens to which the antibodies in the donor patient were directed, but were raised against various nuclear, mitotic spindle, and cytoplasmic antigens. Among the treated groups, animals in the “PBMC” and “PBMC + Bleomycin” groups displayed higher pattern intensities, suggesting increased concentrations of antibodies in the serum (Figure 3).

Despite these findings, the results were not statistically significant and did not correlate with the observed survival outcomes. The diversity of the antibody staining patterns complicates their interpretation, while the elevated antibody levels in the PBMC-treated groups fail to explain the reduced survival in these animals. These observations suggest that, although present, antibodies may not play a direct or dominant role in driving the survival outcomes seen in this model. Instead, other mechanisms, such as inflammation or direct tissue damage, are likely more critical contributors.

### 2.3. Collagen-V-Induced SSc Model Reveals Heart-Specific Fibrotic Changes with Complete Survival

Building on the findings from the bleomycin-induced model, the collagen-V-induced model [12] was employed to evaluate a more immune-focused mechanism of SSc. Unlike the bleomycin model, which is associated with significant mortality and rapid disease progression, the collagen-V-induced model is designed to mimic the autoimmune aspects of SSc without direct chemical insults. This model provides a valuable complementary approach for studying fibrotic and immune-related changes.

Over the 130-day study period, no fatal outcomes or severe symptoms were observed in any treatment group, including the non-treated (NT), adjuvant (Adj), and adjuvant + collagen-V (Adj + Col-V) groups. This starkly contrasts the bleomycin-induced model, highlighting the less acute and less severe nature of the collagen-V-induced model. The complete survival of all animals further emphasizes its utility for studying the autoimmune aspects of SSc in a controlled, chronic setting.

Histological analysis revealed notable organ-specific differences. While hematoxylin-eosin staining (see Appendix A) and Masson’s trichrome staining did not detect observable differences between groups—similarly to the results from the bleomycin model—Picrosirius Red staining identified significant changes in collagen deposition, particularly in the heart. In this organ, the Adj + Col-V treated group demonstrated the highest levels of fibrosis, with significant increases in both thin and thick collagen fibers compared to the NT and Adj groups (Figure 4A,B; Appendix A). In other organs, disruptions in fibrotic materials, including occasional decreases in collagen content, were observed; however, these changes did not reach statistical significance (Appendix A). These observations are consistent with studies showing the localized nature of immune-mediated fibrotic responses in SSc models [12]. These findings reinforce the value of this model for investigating the autoimmune mechanisms underlying SSc and their contribution to localized fibrosis.

### 2.4. Collagen-V-Induced SSc Model Demonstrates Robust Autoantibody Production

The collagen-V-induced model was further evaluated to understand the immune response and autoantibody production over time. Antibody levels against collagen-V were quantified using ELISA, providing a detailed temporal profile of immune activation. A significant increase in anti-collagen-V antibodies was observed in the “Adj. + Col-V” group, confirming the development of a robust immune response targeting human collagen-V (Figure 5A). Antibody production was first detected around day 41 and reached its plateau between days 71 and 130, indicating a progressive and sustained immune activation.

Autoantibody production was confirmed using the HEp-2 IFA, which showed speckled/dotted staining of the cytoplasm in all sera of the “Adj. + Col-V” group from day 41 on, with increasing intensity over time (Figure 5B). This cytoplasmic pattern was very similar to that reported by Teodoro et al. [12] and Callado et al. [18]. Interestingly, the adjuvant-treated group exhibited various patterns (different from the one seen in the “Adj. + Col-V” group) with low to medium intensities, suggesting antibody production without collagen-V immunization. Interestingly, the non-treated group occasionally displayed borderline to low-intensity staining (Figure 5B), which probably represents the baseline xenoreactive antibody repertoire.

Routine serum tests, including damage markers of kidney, muscle, bone, and liver function, showed no significant differences between groups at any time point (Appendix A). These findings suggest that routine biomarkers are unreliable indicators of disease progression or immune activation in this model. Instead, disease monitoring appears to depend on more specific measures, such as anti-collagen-V antibody levels and targeted immune assays.

## 3. Discussion

This study compares the bleomycin-induced and collagen-V-induced models of SSc, highlighting their ability to replicate key disease features, including fibrosis and autoimmunity. The final aim is to better contextualize their significance within the broader framework of SSc research and therapeutic development.

In the bleomycin-induced model, the study replicated key fibrotic processes most prominently observed in the heart, consistent across all bleomycin-treated groups. This consistent cardiac fibrosis indicates that the heart may serve as a reliable marker for fibrosis in this model despite its variable involvement in human SSc. While fibrosis is typically associated with skin and lung involvement in SSc, the localized fibrotic response observed in the heart may reflect unique susceptibilities of organ-specific fibroblasts or ECM composition. This finding aligns with previous research demonstrating that certain organs may exhibit heightened sensitivity to fibrotic stimuli under specific experimental conditions [19].

Interestingly, collagen deposition was reduced in other organs, including the skin and lungs, contrary to the expected accumulation characteristic of SSc. This observation reflects the complex and dynamic nature of fibrosis, which involves both deposition and degradation of collagen. Similar findings have been reported in clinical studies, where fibrosis is not always a linear accumulation of ECM proteins but can involve significant remodeling [20]. This phenomenon complicates the interpretation of fibrosis as a quantitative marker for therapeutic trials, as the increase or decrease in collagen deposition may not reliably correlate with disease severity or progression. Moreover, the route of administration also plays an important role in determining the affected organs. For instance, while subcutaneous administration of bleomycin via osmotic micropump has been shown to induce pulmonary fibrosis, intradermal administration predominantly affects the skin. In contrast, the HOCl model does not typically lead to lung fibrosis, emphasizing its limitations for studying pulmonary involvement in SSc [21]. Future studies could further investigate the molecular mechanisms driving this paradoxical reduction in fibrotic material, including the role of matrix metalloproteinases (MMPs) and other ECM-degrading enzymes [22].

The immune activation observed in the bleomycin model was evidenced by the production of a broad spectrum of antibodies in groups treated with SSc patient-derived PBMCs. The diverse antibody patterns observed in these animals mimic the polyclonal B cell activation seen in human SSc, where autoantibodies target a range of antigens, including nuclear components and ECM proteins [17]. However, the lack of correlation between autoantibody levels and survival outcomes suggests that these immune markers may not directly drive the observed disease severity in this model. Instead, inflammatory cytokine release and direct tissue injury from bleomycin likely play more critical roles in driving fibrosis and mortality. These findings support that while autoantibodies are key diagnostic markers in SSc, their role in disease pathogenesis is often context-dependent, and they may require additional co-factors to exert pathogenic effects [2].

In contrast, the collagen-V-induced model provided a platform for studying antigen-specific autoimmunity, successfully inducing an immune response against collagen-V. Elevated levels of anti-collagen-V antibodies, detected as early as day 41, confirm the model’s utility in replicating immune activation targeting a known autoantigen in SSc. However, this autoimmune response did not lead to widespread fibrosis or significant pathological changes in key organs such as the skin and lungs. Instead, fibrosis was localized to the heart, mirroring findings from the bleomycin model. The limited fibrotic involvement in other tissues may reflect this model’s relatively mild inflammatory environment.

The absence of clinical symptoms and lack of significant fibrosis in the collagen-V model highlight areas for potential refinement. The use of male mice in this experiment may have contributed to the subdued autoimmune response, as females are generally more susceptible to autoimmune diseases due to hormonal influences on immune regulation [13]. Previous studies have shown that using female or older animals can enhance the robustness of autoimmune models, potentially replicating the chronic progression of SSc more effectively [14]. Extending the experimental timeline could allow for the gradual accumulation of fibrosis and autoimmunity, better mimicking the protracted disease course in human SSc.

One of the most striking findings of this study was the consistency of cardiac fibrosis across both models despite their differing mechanisms. This finding suggests that the heart may serve as a uniquely responsive organ in preclinical SSc models, potentially reflecting a convergence of immune-mediated and fibrotic pathways. Although cardiac involvement in SSc is less frequently reported in clinical settings compared to skin and lung involvement, subclinical cardiac fibrosis is increasingly recognized as a significant contributor to morbidity and mortality in SSc patients [23]. Future studies should explore the molecular drivers of cardiac fibrosis in these models, including the role of transforming growth factor β (TGF-β) signaling and fibroblast-specific gene expression [24].

The findings of this project also underscore the need for tailored experimental designs to address each model’s distinct strengths and limitations. The rapid fibrosis and inflammation of the bleomycin model make it ideal for high-throughput screening of antifibrotic agents, such as TGF-β inhibitors or other ECM-targeted therapies [25]. Conversely, the specificity for autoimmune responses in the collagen-V model provides a valuable framework for evaluating immunotherapies, such as chimeric autoantibody receptor (CAAR) T cells [26] targeting collagen-V-reactive B cells.

Combining the strengths of these models could create a hybrid system that more accurately reflects the multifactorial nature of SSc. For example, incorporating collagen-V immunization into the bleomycin model could integrate SSc’s autoimmune and fibrotic components, providing a more comprehensive platform for studying disease progression and therapeutic interventions. Such an approach would align with broader trends in SSc research, which increasingly emphasize the interplay between autoimmunity and fibrosis in driving disease pathogenesis [27].

In conclusion, the findings of this study highlight the complementary strengths of the bleomycin-induced and collagen-V-induced models in replicating distinct aspects of SSc. While neither model fully captures the complexity of human disease, their combined use or further refinement through experimental modifications, such as sex and age adjustments, extended timelines, or hybrid designs, could enhance their translational relevance. By leveraging these insights, future research can advance our understanding of SSc pathophysiology and accelerate the development of targeted therapies for this challenging disease.

## 4. Materials and Methods

### 4.1. Bleomycin-Induced Animal Model

NSG (NOD.Cg-Prkdcscid/Il2rgtm1Wjl/SzJ) mice were purchased from The Jackson Laboratory and housed in a specific-pathogen-free facility under FELASA guidelines and DIN EN ISO 9001 standards [28]. All experiments were approved by the relevant ethical committees, including the National Ethical Committee for Animal Research (#5-1/2017/DEMÁB), and conducted in accordance with the Declaration of Helsinki for the use of human-derived materials.

Peripheral blood mononuclear cells (PBMCs) were freshly isolated from healthy donors and scleroderma patients after obtaining informed consent and IRB approval (RKEB.5378/2019). PBMC isolation was performed using Ficoll-Paque density gradient centrifugation. Isolated PBMCs were resuspended in sterile PBS at a concentration of 5 × 10⁶ cells per injection. NSG mice were engrafted intravenously (i.v.) with 5 × 10⁶ PBMCs in a single injection on day 0. Control groups received no PBMCs (Figure 6A).

Bleomycin was obtained from Teva Pharmaceuticals (Tel Aviv, Israel) and diluted in sterile PBS to a final concentration of 1 mg/mL. Mice were treated subcutaneously (s.c.) with 400 µg bleomycin per injection. Treatments were administered twice weekly for the duration of the experiment. Control groups received either PBMC engraftment or bleomycin treatment alone, while untreated animals served as negative controls (Figure 6A).

Experimental groups included six treatment arms with three mice per group: untreated (NT), PBMCs from healthy donors (PBMC), bleomycin treatment only (BLEO), PBMCs from scleroderma patients (SSc-PBMC), PBMCs from healthy donors combined with bleomycin (PBMC + BLEO), and PBMCs from scleroderma patients combined with bleomycin (SSc-PBMC + BLEO).

Animals were monitored daily for clinical signs of disease progression, including weight loss, skin thickening, and general health status. Humane endpoints were established, and animals were euthanized if predefined criteria were met (Figure 6B).

The animal experiment lasted for 81 days, after which the animals were anesthetized with isoflurane, and retroorbital blood samples were collected into Eppendorf tubes. The animals were then immediately euthanized. The lung, heart, liver, and kidney from each animal were excised, embedded in CryoMatrix gel (Thermo Fisher Scientific, Waltham, MA, USA), snap-frozen in liquid nitrogen-cooled isopentane, and stored at –80 °C for future analysis. Serum was isolated from the collected blood samples via tabletop centrifugation at 21,130 g for 10 min at 4 °C and stored at –80 °C until further use (Figure 6B).

### 4.2. Human Collagen-V-Induced Animal Model

The experimental protocol was adapted from the model described by Teodoro et al. [12] to investigate the effects of collagen-V (ColV) immunization in male C57BL/6 mice. A total of nine mice, aged 8 weeks, were housed in pathogen-free conditions and randomly divided into three treatment groups: (1) “NT” (non-treated), (2) “Adjuvant” (adjuvant-only), and (3) “Adjuvant + ColV” (adjuvant + collagen-V). Human collagen-V (Borstein and Traub Type V or Sigma Type IX; Sigma-Aldrich, St. Louis, MO, USA, C3657-5MG) was dissolved in 10 mM acetic acid to a final 1250 μg/mL concentration. The solution was aliquoted into 300 μL portions, each containing 375 μg of collagen-V, and stored at –20 °C until use. Before injection, the collagen-V solution was emulsified with an equal volume of Freund’s Complete Adjuvant (FCA; Merck/Millipore, Rahway, NJ, USA, 344289-1SET) for initial immunizations or Freund’s Incomplete Adjuvant (FIA; Merck/Millipore, Rahway, NJ, USA, 344291-10ML) for subsequent booster injections (Figure 7A). After the emulsification, 200 μL of the emulsion (containing 125 μg of collagen-V) was injected into each mouse as described below. In the case of the adjuvant-only group, the emulsion was prepared with empty 10 mM acetic acid solution, following the same procedure.

On day 0, all mice in the Adjuvant and Adjuvant + ColV groups received a subcutaneous (s.c.) injection of their respective emulsions prepared with FCA. A booster injection with FCA was administered on day 30. Subsequent booster injections were prepared with FIA and administered intramuscularly (i.m.) on days 45 and 60. The treatment groups were defined as follows: the “NT” group received no treatment throughout the experiment; the “Adjuvant” group received an emulsion of acetic acid solution with FCA or FIA; and the “Adjuvant + ColV” group received an emulsion of collagen-V with FCA or FIA. This immunization schedule was consistent with established protocols for inducing robust immune responses to collagen antigens [12] (Figure 7A).

Blood samples were collected on days 12, 42, 72, and 130. Mice were anesthetized with isoflurane before retroorbital blood collection was obtained through spontaneous voiding. Blood samples were centrifuged at 21,130× *g* for 10 min at 4 °C to isolate serum, which was stored at –80 °C for subsequent analysis. At the end of the experiment, mice were anesthetized with isoflurane and euthanized. Organs were harvested, including the liver, kidney, lung, heart, and shaved dorsal skin. Tissues were embedded in CryoMatrix gel (Thermo Fisher Scientific, Waltham, MA, USA), snap-frozen in liquid nitrogen-cooled isopentane, and stored at –80 °C until further analysis. All procedures were approved by the National Ethical Committee for Animal Research (#5-1/2017/DEMÁB) and conducted in compliance with FELASA and DIN EN ISO 9001 standards (Figure 7B).

### 4.3. Antibody Screening by Indirect Immunofluorescence Assay (IFA)

Serum samples from both animal models were screened for anti-nuclear and anti-cytoplasmic antibodies using indirect immunofluorescence assay (IFA) on HEp-2 cell substrates. The IIFT HEp-20-10 (FA-1522-1010) kit from Euroimmun GmbH (Lübeck, Germany) was utilized following the manufacturer’s instructions with the following protocol modifications to accommodate murine samples (Figure 6B and Figure 7B). For the bleomycin-induced scleroderma model, mouse serum samples were diluted 1:20. In the collagen-V-induced scleroderma model, mouse sera were diluted 1:5 to optimize detection sensitivity. To detect murine antibodies in the collagen-V model, the standard anti-human IgG secondary antibody was substituted with an Alexa Fluor™ 488-labeled goat anti-mouse IgG (H+L) antibody (Thermo Fisher Scientific, Waltham, MA, USA).

Fluorescence patterns were evaluated visually using a fluorescence microscope equipped with an LED light source (Eurostar II Plus, Euroimmun GmbH, Lübeck, Germany). Fluorescence intensity was recorded on a semi-quantitative scale ranging from negative, borderline, 1+, 2+, 3+, to 4+ positive. All observations and scoring were performed independently by a trained investigator to ensure consistency and reliability.

### 4.4. Anti-Collagen-V ELISA

Serum samples from the human collagen-V based model were tested for anti-collagen antibodies in an ELISA assay as in [12], with these alterations: (i) microtiter plates were coated with 1 µg/well collagen-V, (ii) secondary antibody was horseradish peroxidase tagged anti-mouse IgG (Thermo Fisher Scientific, Waltham, MA, USA, 1:10,000 dilution). Enzyme substrate was 3,3′,5,5′-tetramethylbenzidine (BioFX™ TMB One Component HRP Microwell Substrate, cat.: TMBW-1000-01; Surmodics, Minneapolis, MN, USA). Results were expressed as optical density read at 450 nm with a Labsystems Multiskan MS ELISA plate reader (Figure 7B).

### 4.5. Clinical Chemistry Tests

Serum samples were analyzed to assess markers of kidney, muscle, bone, and liver function, indicative of potential organ damage in the experimental models. Measurements were conducted using a Cobas c501 analyzer (Roche, Basel, Switzerland), a well-established system for high-precision biochemical analysis, as reported in similar studies assessing systemic and organ-specific pathologies. The following parameters were measured: urea (mmol/L) as a marker of renal function; total protein (g/L) and albumin (g/L) to evaluate general protein synthesis and nutritional status; creatine kinase (CK) (U/L) to detect muscle damage; alkaline phosphatase (ALP) (U/L) as an indicator of bone or liver involvement; and alanine transaminase (ALT) (U/L) as a marker of hepatic injury. These parameters are routinely used in preclinical and clinical research to monitor systemic and organ-specific health.

### 4.6. Histology

To preserve morphology, tissue samples were washed in physiological saline and fixed in a 4:1 mixture of 40% formaldehyde and absolute ethanol. After fixation, samples were embedded in paraffin, and serial sections of 5 μm thickness were prepared using a microtome. The morphological examination was performed on hematoxylin and eosin (H&E)-stained sections following standard staining protocols (H&E, Sigma-Aldrich, St. Louis, MO, USA). Masson’s Trichrome staining was performed according to the manufacturer’s instructions to visualize collagen fibers (Sigma-Aldrich, St. Louis, MO, USA).

For advanced analysis of collagen fiber orientation and thickness, Picrosirius Red staining (Sigma-Aldrich, St. Louis, MO, USA) was employed. Stained samples were examined under 90° turned polarized light using an Olympus Bx53 polarization microscope (Olympus Corporation, Tokyo, Japan) with a λ/4 compensator. Red coloration in the images indicated thicker collagen fibers, while green coloration corresponded to thinner fibers. Photomicrographs were captured using a DP74 camera (Olympus Corporation, Tokyo, Japan) under constant camera settings and exposure conditions to ensure reproducibility.

Quantitative analysis of collagen fiber thickness and orientation was conducted using ImageJ 1.40g software. The red and green color pixels were measured to evaluate the proportion and distribution of collagen fibers, providing detailed insights into tissue remodeling and fibrosis, consistent with methods described in previous studies evaluating collagen deposition and alignment.

### 4.7. Statistical Analysis

GraphPad Prism 10.4 software (GraphPad Software, Inc., La Jolla, CA, USA) was used for statistical analysis. Data were presented as mean ± SD or SEM. A two-tailed t-test was used to compare two groups. One-way ANOVA with Bonferroni’s post hoc test compared three or more groups. Survival, measured from the time of tumor cell injection, was analyzed by the Kaplan–Meier method and log-rank test. *p* values < 0.05 were considered statistically significant.

## 5. Conclusions

Our study underscores the complementary strengths of the bleomycin-induced and collagen-V-induced models in replicating key aspects of SSc, including fibrosis and autoimmunity. The bleomycin model excelled at mimicking rapid fibrosis, particularly in the heart, but antibody production did not correlate with disease progression, which was very rapid. The collagen-V model effectively replicated antigen-specific autoimmunity but exhibited limited systemic fibrosis, highlighting areas for refinement to improve its clinical relevance. Both models identified cardiac fibrosis as a recurring feature, suggesting the heart’s unique responsiveness to fibrotic and immune-mediated stimuli and positioning the heart as a potential marker of fibrotic activity in mouse preclinical models despite its variable involvement in human SSc.

These findings point to the value of tailored experimental designs to address each model’s limitations while maximizing its translational relevance. The bleomycin model’s rapid fibrosis makes it ideal for antifibrotic therapy screening, while the collagen-V model offers a platform for evaluating immunotherapies targeting antigen-specific immune responses.

Future studies may benefit from hybrid models that integrate the autoimmune and fibrotic components of SSc, reflecting the multifactorial nature of the disease. Refinements such as sex and age adjustments or extended timelines could further enhance the models’ ability to mimic chronic disease progression. Together, these insights provide a foundation for advancing SSc research and developing targeted therapeutic interventions.

## Figures and Tables

**Figure 1 ijms-26-02618-f001:**
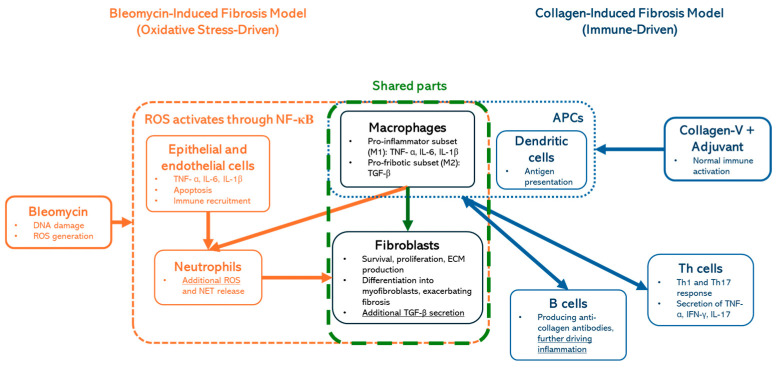
Comparison of bleomycin- and collagen-V-induced fibrosis models: Oxidative stress- vs. immune-driven mechanisms. This figure illustrates the key mechanistic differences and shared components between bleomycin-induced fibrosis (oxidative stress-driven, orange) and collagen-induced fibrosis (immune-driven, blue) models. Macrophages and fibroblasts (green-dashed box) have central roles in fibrotic progression.

**Figure 2 ijms-26-02618-f002:**
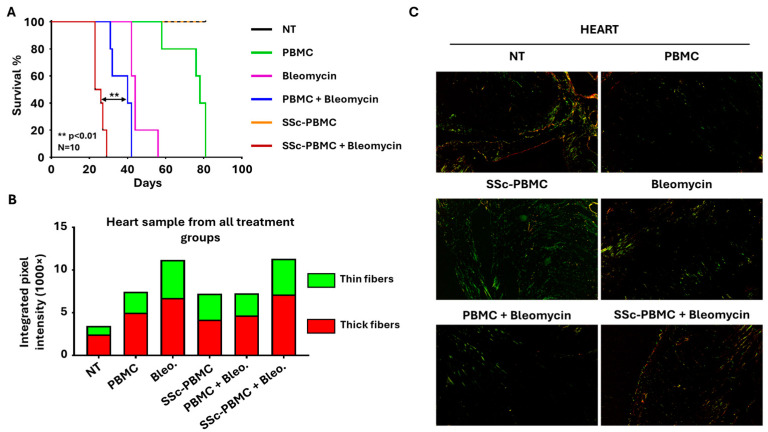
Key outcomes of the bleomycin-induced systemic sclerosis animal model. (**A**) Kaplan–Meier survival analysis demonstrating mortality rates across different treatment groups. The group receiving 400 µg bleomycin with 5 × 10^6^ SSc patient-derived peripheral blood mononuclear cells (SSc-PBMC + Bleomycin; burgundy) exhibited the shortest median survival compared to the group receiving 400 µg bleomycin with 5 × 10^6^ healthy donor-derived PBMCs (PBMC + Bleomycin; blue), 400 µg bleomycin (Bleomycin; magenta), 5 × 10^6^ healthy donor-derived PBMCs (PBMC; green), or 5 × 10^6^ scleroderma patient-derived PBMCs (SSc-PBMC; yellow). SSc-PMBC + Bleomycin vs. PBMC + Bleomycin: ** *p* < 0.01 (log-rank test, N = 10). (**B**) Quantitative analysis of collagen deposition in heart samples, represented by integrated intensities of red and green pixels from Picrosirius Red-stained sections. Red pixels correspond to thick collagen fibers (e.g., type I collagen), while green pixels indicate thinner collagen fibers (e.g., type III collagen. (**C**) Representative Picrosirius Red-stained micrographs of heart tissues from each treatment group, visualized under polarized light. Maginfication was 20×.

**Figure 3 ijms-26-02618-f003:**
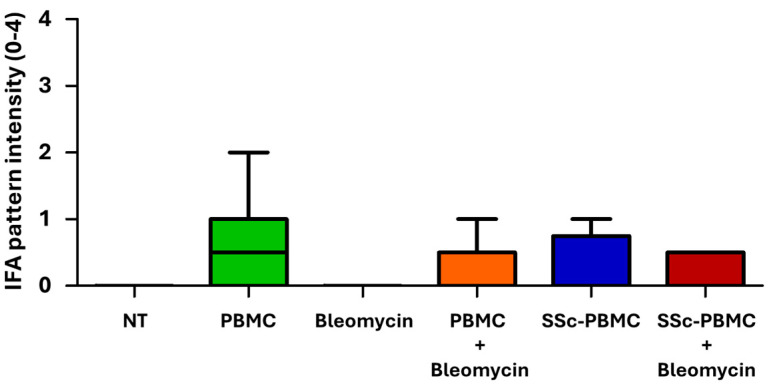
Indirect immunofluorescence assay (IFA) scores of serum samples in the bleomycin-induced model. Antibody levels in serum samples were assessed using indirect immunofluorescence on HEp-2 cells across the treatment groups. The color coding and treatment group designations are consistent with Figure 2. Data are presented as mean scores, with error bars indicating the maximum observed value (n = 3–15 per group). Statistical analysis between PBMC, SSc-PBMC, PBMC + Bleomycin, and SSc-PBMC + Bleomycin groups showed no significant differences in autoantibody levels (*p* > 0.05, non-significant).

**Figure 4 ijms-26-02618-f004:**
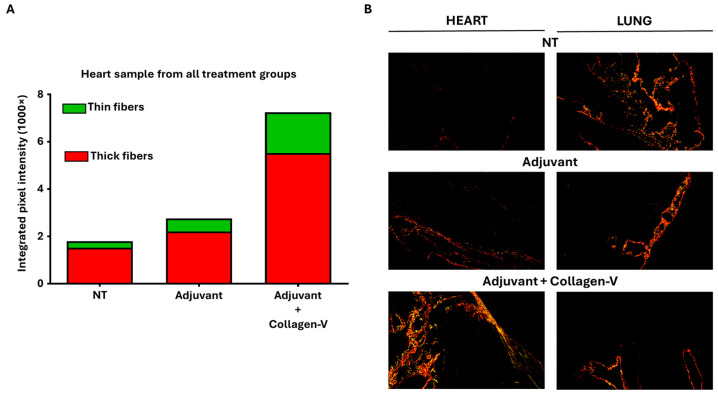
Key outcomes of the human collagen-V-based model. (**A**) Quantitative analysis of collagen content in heart samples from adjuvant + collagen-V cotreated, adjuvant treated and non-treated (NT) groups, represented by integrated intensities of red and green pixels from Picrosirius Red-stained sections. Red pixels correspond to thick collagen fibers (e.g., type I collagen), while green pixels indicate thinner collagen fibers (e.g., type III collagen), as described in fibrosis studies. (**B**) Representative Picrosirius Red-stained micrographs of heart and lung tissues from each treatment group, visualized under polarized light. Maginfication was 20×.

**Figure 5 ijms-26-02618-f005:**
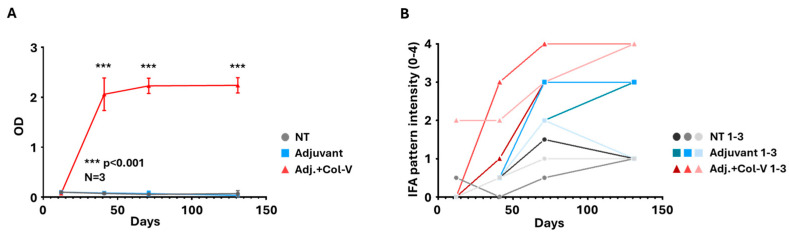
Serum antibody analysis in the human collagen-V-based model. (**A**) Anti-collagen-V antibody levels were measured using ELISA. Data are presented as mean values ± SEM (n = 3 per group). Animals in the adjuvant + collagen-V group (Adj. + Col.-V; red) produced significantly higher levels of anti-collagen-V antibodies compared to the Non-Treated (NT, grey) and Adjuvant-only groups (Adjuvant; blue). Adjuvant + Collagen-V vs. Adjuvant: *** *p* < 0.001. (**B**) Indirect immunofluorescence assay (IFA) scores depicting serum antibody levels assessed on HEp-2 cells. Individual results from each animal (1–3) are shown separately, highlighting variability within treatment groups.

**Figure 6 ijms-26-02618-f006:**
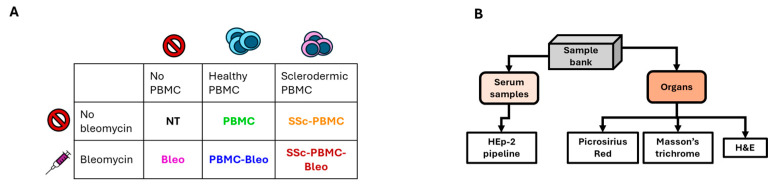
Scheme of the bleomycin-induced animal model. (**A**) Treatment groups. The color coding and treatment group designations are consistent with Figure 2. (**B**) Workflow for the processing of collected serum samples and harvested organs, detailing the steps from collection to analysis.

**Figure 7 ijms-26-02618-f007:**
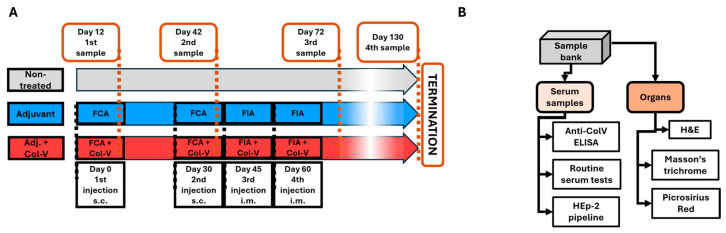
Scheme of the human collagen-V-induced animal model. (**A**) Timeline illustrating the schedule of injections (black frames) and sample collection events (orange frames). Abbreviations: FIA = Freund’s Incomplete Adjuvant; FCA = Freund’s Complete Adjuvant; Col-V = human collagen-V; s.c. = subcutaneous; i.m. = intramuscular. (**B**) Workflow for the processing of collected serum samples and harvested organs, detailing the steps from collection to analysis.

## Data Availability

The data presented in this study are available in this article and its Appendix A.

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
