# Peer review of "Comparative Evaluation of Bleomycin- and Collagen-V-Induced Models of Systemic Sclerosis: Insights into Fibrosis and Autoimmunity for Translational Research"

_ijms, 2025, doi:10.3390/ijms26062618_

Round 1
Reviewer 1 Report
Comments and Suggestions for Authors
Overall, the study is interesting and well-structured. The manuscript is clear in its sections, and the conclusions are relevant to the results.
Introduction, Line 55: The Authors could provide additional details for the readers regarding the action of bleomycin. For example, Bleomycin-induced damage is generally considered to be secondary to oxidative stress and develops in two phases: an initial phase predominantly characterized by tissue inflammation, followed by a phase with a predominance of fibrosis.
Additionally, for completeness, I would also mention a second animal model, the HOCl model, which similarly results from oxidative damage
Discussion: "In the discussion, for completeness, I would point out that the organs and systems involved in the murine model also depend on the route of administration. In particular, the literature describes that bleomycin administered via a subcutaneous micropump can induce pulmonary fibrosis, unlike the HOCl model ( Morozan, A., Joy, S., Fujii, U. et al. Superiority of systemic bleomycin to intradermal HOCl for the study of interstitial lung disease. Sci Rep 13, 20577 (2023))
Author Response
We sincerely appreciate your constructive feedback and the positive assessment of our manuscript. Your insightful comments have helped us refine our work, and we have addressed each of your suggestions in detail below:
Reviewer suggestion 1: Introduction, Line 55: The Authors could provide additional details for the readers regarding the action of bleomycin. For example, Bleomycin-induced damage is generally considered to be secondary to oxidative stress and develops in two phases: an initial phase predominantly characterized by tissue inflammation, followed by a phase with a predominance of fibrosis.
Author’s reply: Thank you for your suggestion to provide further details on bleomycin-induced damage. We have now included a paragraph elaborating on the two-phase mechanism of bleomycin-induced fibrosis.
New paragraph added with 3 new references and a new figure is added:
The tissue damage is primarily driven by oxidative stress, leading to a biphasic re-sponse that mirrors aspects of systemic sclerosis (SSc). The initial inflammatory phase involves immune cell infiltration and the release of cytokines such as TNF-α, IL-6, and TGF-β, contributing to endothelial injury and fibroblast activation (Tashiro et al., 2017). This progresses to a chronic fibrotic phase, characterized by excessive fibroblast prolif-eration, myofibroblast differentiation, and extracellular matrix (ECM) deposition, pri-marily of collagen I and III, leading to progressive tissue stiffening (Herrera et al., 2018). Oxidative stress and persistent cytokine signaling sustain this profibrotic environment, resembling SSc pathogenesis, where vascular and immune dysfunction drive fibrosis in the skin and internal organs (Fernández & Eickelberg, 2012) (Figure1).
Reviewer suggestion 2: Additionally, for completeness, I would also mention a second animal model, the HOCl model, which similarly results from oxidative damage
Author’s reply: We appreciate your suggestion to mention the HOCl model as a complementary oxidative damage model. In response, we have added a new paragraph to the Introduction section to provide a more comprehensive overview of oxidative stress-induced fibrosis models.
New text was added with 2 new references:
While the bleomycin model effectively induces fibrosis, its reliance on direct chemical insult limits its ability to fully replicate SSc pathogenesis. HOCl, a reactive oxygen species produced by myeloperoxidase in neutrophils, triggers endothelial injury, immune activation, and fibroblast proliferation {Mittal et. al, 2014.}. Its systemic administration leads to vascular inflammation, closely mimicking key pathological features of SSc {Maria et al., 2018}. However, fibrosis in this model is less consistent and develops more gradually compared to the bleomycin model, making it less suitable for fibrosis-targeted therapeutic evaluations. Given our focus on well-established fibrotic processes, we prioritized the bleomycin model for its reproducibility and translational relevance.
Reviewer suggestion 3: Discussion: "In the discussion, for completeness, I would point out that the organs and systems involved in the murine model also depend on the route of administration. In particular, the literature describes that bleomycin administered via a subcutaneous micropump can induce pulmonary fibrosis, unlike the HOCl model ( Morozan, A., Joy, S., Fujii, U. et al. Superiority of systemic bleomycin to intradermal HOCl for the study of interstitial lung disease. Sci Rep 13, 20577 (2023))
Author’s reply:Thank you for highlighting the importance of the route of administration in determining organ involvement. We have revised our discussion to include the following:
New text was added with 1 new references:
Moreover, the route of administration also plays an important role in determining the affected organs. For instance, while subcutaneous administration of bleomycin via os-motic micropump has been shown to induce pulmonary fibrosis, intradermal admin-istration predominantly affects the skin. In contrast, the HOCl model does not typically lead to lung fibrosis, emphasizing its limitations for studying pulmonary involvement in SSc {Morozan et al., 2023}.
We hope that these revisions address the reviewer's comments effectively. Thank you once again for your valuable suggestions, which have strengthened our manuscript.
Reviewer 2 Report
Comments and Suggestions for Authors
The authors compare bleomycin-induced and collagen V-induced mouse models of scleroderma.
(1) For bleomycin-induced scleroderma model mice, it is known that the combination with Fli1 model mice induces more pronounced SSc symptoms. This should be clearly stated in the Discussion or Introduction.
Taniguchi T, Asano Y, Akamata K, Noda S, Takahashi T, Ichimura Y, Toyama T, Trojanowska M, Sato S. Fibrosis, vascular activation, and immune abnormalities resembling systemic sclerosis in bleomycin-treated Fli-1-haploinsufficient mice. Arthritis Rheumatol. 2015 Feb;67(2):517-26.
(2) Collagen V-induced scleroderma model mice may not show lung symptoms as the authors mention. On the other hand, the present study compares the bleomycin-induced scleroderma model mice with the collagen V model mice.
Some of what is shown in Supplement (especially skin and lungs) should be shown in the main text.
Author Response
We sincerely appreciate your constructive feedback and the positive assessment of our manuscript. Your insightful comments have helped us refine our work, and we have addressed each of your suggestions in detail below:
Reviewer Suggestion 1: For bleomycin-induced scleroderma model mice, it is known that the combination with Fli1 model mice induces more pronounced SSc symptoms. This should be clearly stated in the Discussion or Introduction.
Author’s reply: We appreciate this insightful suggestion and have now included a discussion of the Fli1-haploinsufficient model in the Introduction section to provide context on how genetic modifications can enhance the bleomycin-induced SSc phenotype. The study by Taniguchi et al. (2015) demonstrated that Fli1-haploinsufficient mice treated with bleomycin exhibit more severe fibrosis, vascular activation, and immune abnormalities, closely resembling human SSc. Given that Fli1 downregulation is implicated in the pathogenesis of SSc, we acknowledge that this combination provides a more robust model for investigating disease mechanisms. We recognize its value in future studies aiming to capture the broader spectrum of SSc pathology.
New text was added with 1 new reference:
To address some of these limitations, genetic modifications have been explored to en-hance the bleomycin-induced SSc phenotype. Fli1-haploinsufficiency has been shown to exacerbate bleomycin-induced fibrosis, leading to a more severe SSc-like phenotype with enhanced vascular dysfunction and immune abnormalities (Taniguchi et al., 2015).
Reviewer suggestion 2: Collagen V-induced scleroderma model mice may not show lung symptoms as the authors mention. On the other hand, the present study compares the bleomycin-induced scleroderma model mice with the collagen V model mice. Some of what is shown in Supplement (especially skin and lungs) should be shown in the main text.
Author’s reply: We agree that providing more direct comparisons of lung pathology between the two models in the main text would improve clarity and completeness. While the Supplementary Figures currently present this data, we recognize that integrating key findings into the main text enhances accessibility for readers. Therefore, we have now incorporated selected histological images from the Supplementary Figures into the Results section. Specifically, we have included representative histological images of lung fibrosis for the collagen V model to illustrate the differences in fibrotic involvement clearly.
We hope that these revisions address the reviewer's comments effectively. Thank you once again for your valuable suggestions, which have strengthened our manuscript.
Reviewer 3 Report
Comments and Suggestions for Authors
Please see attached PDF for detailed comments.

Author Response
We sincerely appreciate your thorough and fair review of our manuscript. Your thoughtful comments and constructive feedback have contributed significantly to improving the clarity and impact of our study. We are particularly grateful for your recognition of the article’s value in assisting readers who are considering implementing this type of model. Our motivation for conducting this research was strongly influenced by real-world challenges, specifically a stalled preclinical trial of a CAAR-T cell therapy targeting SSc due to the inability to identify a suitable animal model. We firmly believe that the coming decade will be defined by breakthroughs in autoimmune disease treatment through next-generation immunotherapies, necessitating humanized systems capable of replicating the role of autoreactive B and T cells and the organ damage these therapies aim to prevent. With this in mind, we are confident that our work will guide researchers to select and refine models that better simulate real-life conditions. Ultimately, such advancements could be the key to developing more effective therapies with fewer side effects.
We have valued and accepted all your suggestions and have incorporated them into the manuscript to the fullest extent possible. Below, we address each of your comments in detail.
Reviewer suggestion 1: Introduction 1.: There is no specific pathway or mechanism highlighted for either method of fibrosis induction and no molecular information that would be useful in an actual hypothesis-driven study. For example, Collagen V fibrosis is mediated by TGF-beta and has downstream targets that would be very useful to know. A table detailing the molecular mechanisms for both pathways is recommended over extensive alterations of the text.
Author’s reply: Thank you for this suggestion. We have now included new text with new references and a new figure summarizing the molecular mechanisms underlying fibrosis induction in both models, including key pathways such as TGF-beta signaling for the Collagen V model and oxidative stress-driven fibroblast activation in the bleomycin model. This addition provides a more precise comparison of the mechanistic basis of each model without overcomplicating the text.
New text was added to the Introduction with new references and a new figure (Figure 1) was also added:
The tissue damage is primarily driven by oxidative stress, leading to a biphasic re-sponse that mirrors systemic sclerosis (SSc) aspects. The initial inflammatory phase involves immune cell infiltration and the release of cytokines such as TNF-α, IL-6, and TGF-β, contributing to endothelial injury and fibroblast activation {Tashiro, 2017 #528}. This progresses to a chronic fibrotic phase, characterized by excessive fibroblast proliferation, myofibroblast differentiation, and extracellular matrix (ECM) deposi-tion, primarily of collagen I and III, leading to progressive tissue stiffening {Herrera, 2018}. Oxidative stress and persistent cytokine signaling sustain this profibrotic environment, resembling SSc pathogenesis, where vascular and immune dysfunction drive fibrosis in the skin and internal organs {Fernandez, 2012} (Figure 1 orange boxes).
It is primarily driven by adaptive immune activation, where dendritic cells (APCs) present collagen antigens, leading to Th1 and Th17 cell responses and the secretion of TNF-α, IFN-γ, and IL-17, which drive inflammation (Figure 1, blue boxes). These mod-els induce the production of autoantibodies that are diagnostic hallmarks of SSc, in-cluding antinuclear antibodies (ANAs) and anti-topoisomerase I antibodies {Cara-maschi, 2015;Henault, 2006 }. While these autoantibodies are pivotal in diag-nosing SSc, their precise roles in disease pathogenesis remain unclear.
Reviewer suggestion 2: Introduction 2.: Although pedantic, the authors should at least mention why both methods aren’t used at the same time in murine studies. Also mention why the Yamamoto method is even bothered with in the current study if it has no immune component. Why would the two methods need to be compared if it’s already known that the Yamamoto method is useless for SSc studies? Wouldn’t a newer or alternative method be better?
Author’s reply: Based on existing literature and our research objectives, we determined that adding an additional autoimmune component to the Bleomycin + SSc PBMC treatment model would not be appropriate, as it could further accelerate the already rapid mortality observed in these animals. This could obscure our ability to assess long-term disease progression and therapeutic interventions. Additionally, from an animal welfare perspective, performing a joint titration of both systems posed significant challenges. There is currently no established empirical evidence to determine a minimal bleomycin dose that would allow for prolonged survival while ensuring that the autoreactive effects induced by Collagen V remain the dominant pathogenic factor. Instead, our findings, in agreement with previous studies, suggest that the Collagen V induction model provides a more promising approach for replicating key aspects of SSc pathogenesis, particularly in capturing the role of autoimmunity in disease progression. Aligned with emerging research trends, we are actively optimizing this model by increasing the frequency of treatments within a shorter timeframe, aiming to enhance disease progression in a controlled manner. In parallel, we are investigating the efficacy of mouse-derived CAAR-T cells in modulating autoantibody production, a crucial step in developing targeted immunotherapies for autoimmune diseases like SSc. By refining the experimental parameters and incorporating novel therapeutic testing, we aim to develop a more physiologically relevant model that can serve as a foundation for studying SSc mechanisms and evaluating next-generation immunotherapies with greater translational potential.
Reviewer suggestion 3: Results 1.: As expected with a well-studied system, the authors were able to successfully induce SSc by following a protocol. It is excellent to have a model system that reliably works.
Author’s reply: We sincerely appreciate your kind words and recognition of the reliability of our model system. Our primary goal was to provide a clear and reliable foundation that other researchers can build upon, and your feedback reassures us that we have achieved this objective. Thank you for your encouraging comment!
Reviewer suggestion 4: Results 2.: Lines 134-136: This is the first time this aim has been mentioned as a primary aim. In the Introduction, it seems as if it’s secondary only. It should be a primary aim and emphasized as such.
Author’s reply: We appreciate the reviewer’s insightful suggestion and have added new text to the Introduction to emphasize the HEp-2 IFA experiment as a primary aim. This revision clearly conveys its significance in assessing autoantibody involvement in disease progression and survival outcomes.
New text was added to the manuscript with one new reference:
The assessment of antinuclear (ANA) and anticytoplasmic autoantibodies using the HEp-2 IFA experiment was a key focus of our study, given the well-established role of autoantibodies in systemic sclerosis (SSc) pathogenesis. Autoantibodies such as anti-topoisomerase I, anti-centromere, and anti-RNA polymerase III have been strongly associated with disease severity, specific clinical manifestations, and survival outcomes in SSc patients {Mecoli et al., 2018. Given these clinical implications, our study aimed to determine whether ANA and anti cytoplasmic autoantibodies influence disease progression and survival outcomes in our murine models.
Reviewer suggestion 5: Results 3.: Not altering the modification or testing of IgG-producing cells to see the effect on pathogenesis is an obvious oversight, but understandable since the results are not statistically significant. However, in Line 153, the implication of direct tissue damage as a more critical contributor hints that the Yamamoto method is superior as the Introduction explicitly states chemical injury as the main factor in pathogenesis.
Author’s reply: Our goal in this study was to establish a model for a CAAR-T cell system, in which genetically engineered effector cells specifically eliminate autoantibody-inducing B cell populations. When we observed that the bleomycin-induced model did not result in significant autoantibody production, we did not further investigate the background of this phenomenon, as it was irrelevant to our specific model. We fully acknowledge the reviewer's insightful perspective that a more detailed exploration of this could have provided more profound insight into the mechanisms involved. Future studies incorporating both autoantibody modulation and fibrotic injury could help elucidate the precise role of IgG-producing cells in systemic sclerosis pathogenesis.
Reviewer suggestion 6: Results 4.: Line 172-174: Is the survival of all animals expected and a part of the protocol? Is this reflective of real SSc? Likewise, in Lines 205-219, much is made of high autoantibody production but then organ damage markers are insignificant between the two test groups. This calls into question the utility of a disease model that does not model the disease.
Author’s reply: While we initially hypothesized that autoantibody production would lead to tissue damage severe enough to impact survival, this was not observed in our base model. However, this does not suggest that the approach itself is unsuitable for simulating SSc. Our findings clearly demonstrate the production of disease-relevant autoantibodies, but it is possible that the exposure duration was insufficient to induce significant pathological effects. In our ongoing research, as referenced in reply to “Reviewer Suggestion 2: Introduction 2”, we are testing a more extended model where individuals are exposed for longer periods and at higher frequencies. In this refined system, we aim to evaluate the efficacy of CAAR-T cells, using changes in specific autoantibody production as a primary outcome while also considering survival as a secondary marker of disease progression.
Reviewer suggestion 7: Discussion 1.: Not surprisingly, there are benefits and drawbacks to both methods. However, there is a lack of comparative reports using either or both systems and the analysis of the differences. Please add some other studies that offer contrast to the value of the disease models, especially since no real significant results were obtained.
Author’s reply: We appreciate the reviewer's suggestion and fully recognize the importance of including further comparative studies to highlight the value of different SSc disease models.
The HOCl-induced model, as described by Servettaz et al. (Servettaz A, Goulvestre C, Kavian N, Nicco C, Guilpain P, Chereau C, et al. Selective oxidation of DNA topoisomerase 1 induces systemic sclerosis in the mouse. J Immunol 2009; 182: 5855–64.), has been demonstrated to mimic better the vascular and immune dysfunction observed in SSc patients. This model effectively replicates critical features of SSc pathogenesis, such as endothelial damage and immune activation, thereby offering a complementary approach to fibrosis-dominated models. Moreover, it is also underscored the central role of oxidative stress in fibrosis development across different SSc models. Their findings reinforce the necessity of careful model selection based on specific research objectives, as different models emphasize distinct aspects of disease progression.
Given the significance of model selection in preclinical studies, we have now incorporated relevant text and references into both the Introduction and Discussion to contextualize our findings within the broader landscape of SSc research. By doing so, we aim to provide a more comprehensive understanding of how different models contribute to our knowledge of SSc pathogenesis and therapeutic development. We hope this addition allows for a more robust discussion of the strengths and limitations of our approach while offering valuable insights for future studies seeking to refine disease modeling strategies.
New text was added to the Introduction with 2 new references:
While the bleomycin model effectively induces fibrosis, its reliance on direct chemical insult limits its ability to fully replicate SSc pathogenesis. HOCl, a reactive oxygen species produced by myeloperoxidase in neutrophils, triggers endothelial injury, immune activation, and fibroblast proliferation {Mittal et. al, 2014.}. Its systemic administration leads to vascular inflammation, closely mimicking key pathological features of SSc {Maria et al., 2018}. However, fibrosis in this model is less consistent and develops more gradually compared to the bleomycin model, making it less suitable for fibrosis-targeted therapeutic evaluations. Given our focus on well-established fibrotic processes, we prioritized the bleomycin model for its reproducibility and translational relevance.
New text was added to the Discussion with 1 new references:
Moreover, the route of administration also plays an important role in determining the affected organs. For instance, while subcutaneous administration of bleomycin via os-motic micropump has been shown to induce pulmonary fibrosis, intradermal admin-istration predominantly affects the skin. In contrast, the HOCl model does not typically lead to lung fibrosis, emphasizing its limitations for studying pulmonary involvement in SSc {Morozan et al., 2023}.
Reviewer suggestion 8: Discussion 2.: Line 274-281: If it is well known that female mice are better, why were male mice chosen?
Author’s reply: We appreciate this important point about the selection of male mice in our study. Initially, the bleomycin-induced fibrosis model was performed using female mice and cell preparations derived from human female donors. Given the unexpectedly strong effect observed in this experiment and the associated ultrashort survival, we were hesitant to use male mice in the subsequent Collagen-V model, which was initiated later. Additionally, we hypothesized that using male mice might reduce variability due to hormonal fluctuations, which are known to influence immune system activity and fibrosis progression. By minimizing this potential confounding factor, we aimed to enhance the reproducibility and comparability of our results. However, based on the findings presented in this paper and growing evidence supporting the use of female mice in the Collagen-V model. Thus, we have now adopted female individuals for our ongoing preclinical study, which was mentioned above.
Reviewer suggestion 9: Methods 1. and 2.: The methods seem appropriate and well written. The detail in the methods is welcome for those readers who wish to use one or the other disease models.
Author’s reply:
We sincerely appreciate the reviewer's recognition of the methodological rigor applied in this study. Ensuring the robustness and clarity of our methods was a key objective, and we are pleased that our efforts have been acknowledged.
We sincerely appreciate your thoughtful review and valuable suggestions, which have significantly improved our manuscript. By addressing these points, we believe we have enhanced the clarity, depth, and applicability of our study, making it a more useful resource for researchers in the field. Thank you again for your time and expertise.
Round 2
Reviewer 2 Report
Comments and Suggestions for Authors
The authors have made sufficient revisions.